# The Effect of Previous Exposure to Malaria Infection and Clinical Malaria Episodes on the Immune Response to the Two-Dose Ad26.ZEBOV, MVA-BN-Filo Ebola Vaccine Regimen

**DOI:** 10.3390/vaccines11081317

**Published:** 2023-08-02

**Authors:** Daniela Manno, Catriona Patterson, Abdoulie Drammeh, Kevin Tetteh, Mattu Tehtor Kroma, Godfrey Tuda Otieno, Bolarinde Joseph Lawal, Seyi Soremekun, Philip Ayieko, Auguste Gaddah, Abu Bakarr Kamara, Frank Baiden, Muhammed Olanrewaju Afolabi, Daniel Tindanbil, Kwabena Owusu-Kyei, David Ishola, Gibrilla Fadlu Deen, Babajide Keshinro, Yusupha Njie, Mohamed Samai, Brett Lowe, Cynthia Robinson, Bailah Leigh, Chris Drakeley, Brian Greenwood, Deborah Watson-Jones

**Affiliations:** 1London School of Hygiene and Tropical Medicine, London WC1E 7HT, UK; 2EBOVAC Project Office, Kukuna Road, Kambia, Sierra Leone; 3College of Medicine and Allied Health Sciences, University of Sierra Leone, New England Ville, Freetown, Sierra Leone; 4Mwanza Intervention Trials Unit, National Institute for Medical Research, Mwanza P.O. Box 11936, Tanzania; 5Janssen Research and Development, 2340 Beerse, Belgium; 6Janssen Vaccines and Prevention, 2333 CB Leiden, The Netherlands; 7KEMRI-Wellcome Trust Research Programme, Kilifi P.O. Box 230, Kenya

**Keywords:** malaria, Ebola, vaccine, immunogenicity, Ad26.ZEBOV, MVA-BN-Filo

## Abstract

We assessed whether the immunogenicity of the two-dose Ad26.ZEBOV, MVA-BN-Filo Ebola vaccine regimen with a 56-day interval between doses was affected by exposure to malaria before dose 1 vaccination and by clinical episodes of malaria in the period immediately after dose 1 and after dose 2 vaccinations. Previous malaria exposure in participants in an Ebola vaccine trial in Sierra Leone (ClinicalTrials.gov: NCT02509494) was classified as low, intermediate, and high according to their antibody responses to a panel of *Plasmodium falciparum* antigens detected using a Luminex MAGPIX platform. Clinical malaria episodes after vaccinations were recorded as part of the trial safety monitoring. Binding antibody responses against the Ebola virus (EBOV) glycoprotein (GP) were measured 57 days post dose 1 and 21 days post dose 2 by ELISA and summarized as Geometric Mean Concentrations (GMCs). Geometric Mean Ratios (GMRs) were used to compare groups with different levels of exposure to malaria. Overall, 587 participants, comprising 188 (32%) adults (aged ≥ 18 years) and 399 (68%) children (aged 1–3, 4–11, and 12–17 years), were included in the analysis. There was no evidence that the anti-EBOV-GP antibody GMCs post dose 1 and post dose 2 differed between categories of previous malaria exposure. There was weak evidence that the GMC at 57 days post dose 1 was lower in participants who had had at least one episode of clinical malaria post dose 1 compared to participants with no diagnosed clinical malaria in the same period (GMR = 0.82, 95% CI: 0.69–0.98, *p*-value = 0.02). However, GMC post dose 2 was not reduced in participants who experienced clinical malaria post-dose 1 and/or post-dose 2 vaccinations. In conclusion, the Ad26.ZEBOV, MVA-BN-Filo Ebola vaccine regimen is immunogenic in individuals with previous exposure to malaria and in those who experience clinical malaria after vaccination. This vaccine regimen is suitable for prophylaxis against Ebola virus disease in malaria-endemic regions.

## 1. Introduction

Ebola disease indicates a group of severe, often fatal, infections caused by viruses of the genus *Orthoebolavirus* [1,2]. In 2018, the World Health Organization (WHO) included Ebola disease among the infections that pose the greatest public health risk due to their epidemic potential and for which vaccine development is urgently needed [3]. Since then, two vaccine regimens against Ebola virus (EBOV), species *Zaire ebolavirus*, which causes Ebola virus disease (EVD) [4], have reached a more advanced stage of clinical development. A single-dose recombinant vesicular stomatitis virus-vectored vaccine expressing the EBOV glycoprotein (rVSV-ZEBOV-GP, Ervebo) has been licensed for use in adults in the EU, USA, and various African countries [5,6,7] and has received WHO pre-qualification [8]. Similarly, a heterologous two-dose regimen, consisting of an adenovirus type 26 (Ad26) vector-based vaccine encoding the EBOV glycoprotein (Ad26.ZEBOV, Zabdeno) and the modified vaccinia Ankara (MVA) vector-based vaccine, encoding EBOV, Sudan virus and Marburg virus glycoproteins, and the Taï Forest virus nucleoprotein (MVA-BN-Filo, Mvabea), has obtained conditional marketing authorization in the EU and various African countries and WHO pre-qualification for prophylactic use in adults and children aged 1 year or older [9,10,11].

All EVD outbreaks have occurred in sub-Saharan Africa [4] and vaccination against EVD will likely be implemented most frequently in countries in this region. Sub-Saharan Africa also has the highest burden of malaria in the world. In 2021, the WHO estimated that 95% of all malaria cases and 96% of all malaria deaths occurred in this region [12]. It is therefore important to evaluate whether malaria can affect the immune responses after vaccination against EVD.

The effect of malaria on reducing the immune response to some vaccines is well recognized. Impaired humoral responses have been observed in children with symptomatic malaria following vaccination with tetanus toxoid and typhoid [13], *Hemophilus influenzae* type b conjugate vaccine [14], and meningococcal polysaccharide vaccine [15]. In a study in Nigeria, children receiving malaria chemoprophylaxis showed higher antibody responses to a meningococcal polysaccharide vaccine than children of the same age who did not receive malaria chemoprophylaxis [15]. However, malaria infection did not impair the immune responses to the prophylactic HPV-16/18 virus-like particle, AS04-adjuvanted vaccine in adolescents and adults in East Africa [16]. A study in adults who received the rVSV-ZEBOV-GP Ebola vaccine found no evidence of an impaired immune response in participants with asymptomatic malaria infection at vaccination [17]. Another study in adults and children who received the Ad26.ZEBOV, MVA-BN-Filo Ebola vaccine regimen found that anti-EBOV-GP antibody concentrations after dose 1 and before dose 2 were lower in 1–3-year-old children with asymptomatic malaria infection at vaccination compared with malaria-negative children of similar age [18]. However, antibody concentrations after dose 2, a measure of the overall immunogenicity of the vaccine regimen, were not significantly different between the two groups [18]. This study also found no consistent effect of asymptomatic malaria infection on vaccine-induced immune responses across other age groups.

In this analysis of participants from the same study [18], we evaluated whether the immunogenicity of the two-dose Ad26.ZEBOV, MVA-BN-Filo Ebola vaccine regimen was affected by exposure to malaria before dose 1 vaccination and by clinical episodes of malaria in the period immediately after dose 1 or dose 2 vaccinations. To assess previous exposure to malaria, we measured antibodies to a panel of six *Plasmodium falciparum* (*P. falciparum*) recombinant antigens with a Luminex MAGPIX quantitative suspension array technology (qSAT) platform [19]. This method allows the evaluation of both long-term and recent malaria exposure and has been previously validated in children with malaria infection [20].

## 2. Materials and Methods

### 2.1. Study Design and Participants

A nested cohort study on malaria was implemented in the EBOVAC-Salone trial, which investigated the safety and immunogenicity of the two-dose regimen with the Ad26.ZEBOV and MVA-BN-Filo Ebola vaccines in adults and children in Sierra Leone. This is the same cohort of participants in which we previously examined the effect of asymptomatic malaria parasitemia at the time of vaccination on the vaccine-induced immune responses [18].

The EBOVAC-Salone trial occurred between September 2015 and July 2018 in Kambia District, an area in the North of Sierra Leone affected by intense malaria transmission [21]. Information on the design of the clinical trial can be found on the trial registration page in ClinicalTrials.gov (NCT02509494) and has been described previously [22]. The trial was conducted in two stages: Stage 1 in which a small number of adults were all vaccinated with Ad26.ZEBOV followed by MVA-BN-Filo after 56 days, and Stage 2 in which adults and children were randomized to receive either the same Ebola vaccine regimen as in Stage 1 or an active control vaccine [22]. All participants in Stage 2 of the EBOVAC-Salone trial, which included 400 adults (≥18 years of age) and 576 children in three age cohorts (1–3 years, 4–11 years, and 12–17 years), were offered the opportunity to take part in the malaria cohort study on the day of dose 1 vaccination. A separate informed consent process from the one used for the main trial was implemented to obtain informed consent (including assent in children aged 7–17 years) for the malaria study. Only individuals who received the Ebola vaccine regimen in accordance with the trial protocol and gave their consent to the malaria study were included in the analysis after the trial was over and the data were unblinded. Additional information on the design of the malaria cohort study has been presented in a previous publication [18].

### 2.2. Assessment of Exposure to Malaria Infection

Exposure to malaria infection was categorized in three ways: (1) exposure to malaria before vaccination, (2) exposure to malaria at vaccination (described in a previous publication [18]) and (3) exposure to malaria after vaccination but before assessment of vaccine immunogenicity (Figure 1).

To assess prior exposure to malaria, we categorized participants based on their antibody responses to a panel of *P. falciparum* antigens, using serum samples collected for an EBOV seroprevalence study [23] at the screening visit of the EBOVAC-Salone trial, which occurred within 28 days before the administration of dose 1 vaccination. Since the sample identification numbers for the seroprevalence study were unlinked from the EBOVAC-Salone trial and the malaria cohort study identification numbers, a matching algorithm based on participants’ date of birth, age, sex, date of screening, and study clinic number was used to link the seroprevalence study samples to the participants included in the malaria cohort study. Only participants with an exclusive one-to-one matching, which allowed the identification of an available serum sample for the laboratory analysis, were included in this analysis. The data linkage and the use of samples were approved by both the Sierra Leone Ethics Committee and the London School of Hygiene and Tropical Medicine Ethics Committee.

Samples were analyzed using the Luminex MAGPIX qSAT platform as described in previous publications [19]. A brief description of the technique is also provided in the Appendix A. In a previous study by Achan et al. [19], which employed the same technique, antibody responses to the following six *P. falciparum* antigens were considered the most appropriate to determine exposure to malaria infection: apical membrane antigen 1 (AMA-1), merozoite surface protein 1.19 (MSP-1.19), and glutamate-rich protein (GLURP.R2) reflecting long-term exposure to malaria; reticulocyte-binding protein homologue (Rh2.2030), gametocyte exported protein (GEXP18), and early transcribed membrane protein (Etramp5.Ag1) reflecting recent exposure to malaria (i.e., infection in the past ~9 months). Antibody responses to each antigen are expressed as Median Fluorescent Intensity (MFI) [19]. Participants were ranked into quartiles of their MFI to each antigen within their age group and were given a score from 1 to 4 according to the quartile to which they belonged (1 for the lowest quartile, 4 for the highest quartile). The scores obtained for each antigen were added to obtain an age-adjusted cumulative quartile score. Participants were then categorized according to the quartiles of this score. The highest (4th quartile) and lowest (1st quartile) scores were assigned to high-exposure and low-exposure groups, respectively [19]. Participants with intermediate scores (2nd and 3rd quartiles) were assigned to the intermediate-exposure group [19]. We considered participants in the high-exposure group to have the highest exposure to malaria infection, while participants in the low-exposure group had the lowest exposure to malaria infection. Participants in the intermediate-exposure group were considered to have intermediate exposure to malaria infection. Since long-term and recent exposure to malaria might have a different effect on vaccine-induced immune responses, we conducted a secondary analysis of the data, ranking participants separately for their responses to the long-term exposure antigens (AMA-1, MSP-1.19, GLURP.R2) and antigens indicative of recent malaria infection (Rh2.2030, GEXP18, Etramp5.Ag1).

To assess the potential impact of malaria after vaccination, we considered episodes of symptomatic malaria recorded after vaccination and before the assessment of immunogenicity. As part of the safety monitoring in the EBOVAC-Salone clinical trial, adverse events (AEs) were collected for 28 days after each vaccine dose, while serious adverse events (SAEs) were collected throughout the study. Malaria was the most frequent unsolicited AE after each vaccination in all age groups and the most frequent SAE in 1–3-year-old children [22]. Diagnosis of malaria was based on clinical symptoms and positivity to the First Response Malaria Ag. pLDH HRP2 Combo Rapid Diagnostic Test (Premier Medical Corporation Private Limited, Mumbai). Participants with clinical malaria received a complete course of age-appropriate antimalarial drugs according to national malaria treatment guidelines [24].

### 2.3. Assessment of Vaccine-Induced Immune Responses

Serum samples were obtained at baseline on Day 1 (immediately before dose 1 vaccination), on Day 57 (immediately before dose 2), and on Day 78 (21 days post dose 2) for the assessment of IgG antibodies to EBOV glycoprotein (GP), using the Filovirus Animal Non-Clinical Group (FANG) ELISA at Q2 Solutions Vaccine Testing Laboratory in the USA (https://www.q2labsolutions.com, accessed on 31 July 2023). The FANG ELISA was validated by the USA Food and Drug Administration (FDA) in February 2017 (Q2 Solutions, pers. comm., 2017).

### 2.4. Sample Size Calculation and Statistical Analysis

We performed a lognormal power calculation and determined that 460 participants with an available serum sample at screening, divided into quartiles for previous exposure to malaria, would allow a comparison of approximately 115 high-exposure participants with 115 low-exposure participants giving 90% power to detect a geometric mean ratio (GMR) of 0.81 and 80% power to detect a GMR of 0.83 for EBOV GP binding antibodies in high-exposure participants compared with low-exposure participants, assuming a coefficient of variation of 0.5 within each of the two groups, and an alpha (Type I) error of 0.05. For the assessment of the effect of clinical malaria episodes after vaccination on vaccine immunogenicity, we assumed that at least 117 (20%) of the 587 participants would have had at least one episode of malaria post-dose 1 vaccination, based on data from the EBOVAC-Salone trial [22]. This gave us 90% power to detect a GMR of 0.81 and 80% power to detect a GMR of 0.83 for EBOV GP binding antibodies in participants with no malaria episodes compared to those with at least one episode, using the same lognormal power calculation and assumptions as before.

Demographic characteristics were summarized using descriptive statistics. An age-adjusted cumulative quartile score and categories of previous exposure to malaria were obtained from the MFI for the six *P. falciparum* antigens as described previously. Malaria symptomatic infections after vaccination were analyzed as a categorical variable (i.e., at least one episode of malaria) and as a continuous variable (i.e., number of episodes). The measurements of EBOV GP binding antibody concentration were transformed on the logarithmic scale and summarized as geometric mean concentration (GMC) with 95% confidence interval (CI). Mean log-transformed EBOV GP antibody concentrations were compared between exposure categories, overall, and in each age group, using linear regression. The regression coefficients and 95% CI were back-transformed to obtain GMR, i.e., ratios of GMCs between exposure categories, with 95% CI. Statistical evidence of a difference in antibody GMC between different categories was assessed using a log likelihood-ratio test. We considered age as an a priori confounder. For this reason, we used age-adjusted categories to assess the effect of previous exposure to malaria on vaccine immunogenicity, while the effect of clinical episodes of malaria after vaccination was adjusted for age by including age group as an independent variable in the linear regression model. The analysis of the effect of previous exposure to malaria on vaccine immunogenicity was also adjusted for baseline EBOV GP-specific antibody concentrations pre-dose 1 vaccination because this variable was considered a potential confounder. This was achieved by including the baseline EBOV GP-specific antibody concentrations as an independent variable in the linear regression model.

## 3. Results

### 3.1. Study Participants

Among 976 participants enrolled in Stage 2 of the EBOVAC-Salone trial, 730 were randomized to the Ebola vaccine group and received the first dose of this regimen (Figure 2).

Among these participants, 140 were not eligible for inclusion in the malaria ancillary study, consisting of 55 who did not receive dose 2, 72 who received a delayed dose 2 beyond the protocol visit window due to a temporary halt of the trial, three who did not consent to the malaria study, and 10 who were not eligible for other reasons (Figure 2). For three participants, the vaccine immunogenicity data were not available. The remaining 587 participants were included in the malaria study. Of those, 188 (32.0%) were adults aged ≥ 18 years, and 399 (68.0%) were children (125 were aged 1–3 years, 133 aged 4–11 years, and 141 aged 12–17 years). The demographic characteristics of the study participants have been published previously [18] and are presented in the Appendix A.

### 3.2. The Effect of Previous Exposure to Malaria on Vaccine Immunogenicity

Overall, 474 (80.7%) of 587 study participants had an available serum sample collected at screening, which allowed the assessment of previous exposure to malaria before vaccination (Figure 2). When the EBOV GP–specific antibody GMC after each vaccine dose was compared between categories of previous exposure to malaria at screening in all participants and then stratified by age cohort (Table 1 and Table 2), there was no evidence that the EBOV GP binding antibody GMC post dose 1 (Day 57) and post dose 2 (Day 78) differed between different categories of previous exposure to malaria. The analysis in which participants were ranked separately for their responses to the long-term exposure antigens and antigens indicative of recent malaria infection showed similar results (Appendix A).

### 3.3. The Effect of Clinical Episodes of Malaria after Vaccination on Vaccine Immunogenicity

Among the 587 participants in the malaria study, 175 (29.8%) had at least one episode of clinical malaria recorded between Day 1 and Day 57, when the immunogenicity of dose 1 was assessed (Table 3). When considering participants overall and after adjusting for age group, there was weak evidence that participants who had at least one episode of clinical malaria post dose 1 had a lower GMC at Day 57 compared to participants who had no recorded episodes of clinical malaria (GMR = 0.82, 95% CI: 0.69–0.98, *p*-value = 0.02), (Table 3). After stratifying for age group, there was also weak evidence that adults who had at least one episode of clinical malaria had a lower GMC at Day 57 than adults with no recorded episodes of malaria (age group–specific GMR = 0.76, 95% CI: 0.57–1.00, *p*-value = 0.05), (Table 3). In the other age groups, there was no evidence of a difference, although the GMCs tended to be lower in participants with at least one episode of clinical malaria in the 1–3 and 4–11 year old groups (Table 3).

Among 579 participants who received dose 2 and had immunogenicity results available 21 days post dose 2 (Day 78), 229 (39.5%) had at least one episode of malaria recorded since dose 1 vaccination, in between Day 1 and Day 78 (Table 4). When considering participants overall and after adjusting for age group, there was no evidence of a difference in EBOV GP-specific binding antibody GMC between participants who had at least one episode of clinical malaria compared to participants who had no clinical malaria (Table 4). The result was similar when participants were stratified by age group (Table 4) or when only post-dose 2 malaria episodes, recorded between Day 57 and Day 78, were considered (Table 5).

Among the 229 who had at least one episode of malaria recorded between Day 1 and Day 78, 177 (77.3%) had only one episode, 49 (21.4%) had two episodes, and 3 (1.3%) had three episodes. There was no evidence of an association between the number of episodes of malaria and the binding antibody concentration at 21 days post dose 2 (GMR = 0.97, 95%CI = 0.85–1.11, *p*-value = 0.65).

## 4. Discussion

Countries in Sub-Saharan Africa that are considered at risk of future EVD outbreaks also have a high burden of malaria [4,12,25]. This means that vaccination against EVD will be implemented most frequently in areas with a high malaria prevalence. Since malaria is recognized as having an effect on the immune responses of some vaccines [13,14,15], we conducted a study to assess whether malaria impaired the immunogenicity of the two-dose Ad26.ZEBOV, MVA-BN-Filo Ebola vaccine regimen in adults and children from Kambia district in Sierra Leone, an area where malaria is highly prevalent [18,21].

The results presented in this paper show no evidence of a difference in EBOV GP-specific binding antibody concentrations between categories of previous exposure to malaria infection based on antibody responses to six *P. falciparum* antigens indicative of long-term exposure (AMA-1, MSP-1.19 and GLURP.R2) and recent exposure, i.e., infection in the past ~9 months, (Rh2.2030, GEXP18 and Etramp5.Ag1) to malaria. Participants who had at least one episode of symptomatic malaria after dose 1 had lower antibody concentrations 57 days after dose 1 compared with participants with no malaria. However, when we assessed the cumulative effect of malaria episodes post dose 1 and post dose 2 on the antibody concentrations 21 days after dose 2, we did not observe any evidence of a difference.

These results could be explained by transient suppression of heterologous antibody production during clinical malaria episodes. Malaria is known to dysregulate B-cell functions, which could affect the production of antibodies [13]. These results are consistent with previous results from the same study [18], which showed that young children (1–3 years old) who had asymptomatic malaria infection at dose 1 (Ad26.ZEBOV) vaccination had lower antibody concentration post dose 1 compared with malaria-negative children (age group–specific GMR = 0.56; 95% CI: 0.39–0.81), but this was not observed in older age groups post dose 1 or across all age groups post dose 2.

The fact that we did not observe any effect of previous exposure to malaria on vaccine-induced immune responses, while we observed some effect of asymptomatic malaria infection at vaccination and of clinical malaria after vaccination, could be due to the different time intervals between these infections and the evaluation of vaccine immunogenicity. A recent malaria infection is probably more likely to have a detectable effect on vaccine immunogenicity than an infection that occurred in the past. However, this result could also be due to the way we assessed these exposures. Previous exposure to malaria at screening was based on the antibody response to malaria antigens, and it is possible that participants with higher levels of antibodies to malaria might also be better at producing an antibody response after vaccination against EVD. Asymptomatic malaria infection assessed through microscopy [18] or clinical episodes of malaria recorded in the clinical trial and confirmed with a positive RDT might have been a better way to capture exposure to malaria infection.

However, the current and the previous analysis [18] show that the effect of malaria infection on vaccine immunogenicity was only detected post dose 1 at Day 57, while no effect was observed at 21 days post dose 2 when the immunogenicity of the two-dose Ebola vaccine regimen was primarily evaluated in the EBOVAC-Salone trial [22]. Thus, even if participants with asymptomatic malaria infection at the time of vaccination or clinical malaria after vaccination have lower antibody responses post dose 1, they respond after the administration of dose 2 and produce antibody concentrations that are similar to those observed in participants not affected by malaria [18].

These findings are also consistent with those from another study conducted in Sierra Leone [17], which showed robust immune responses to another Ebola vaccine (rVSVΔG-ZEBOV-GP) in asymptomatic adults with malaria parasitemia at vaccination, and this is reassuring because it confirms that both vaccines are immunogenic in malaria-endemic areas. rVSVΔG-ZEBOV-GP was also shown to be effective in preventing EVD in Guinea, a malaria-endemic country [26,27].

Our analysis has some limitations. Serum samples were no longer available from the EBOVAC-Salone trial and, therefore, serum samples from an ancillary EBOV seroprevalence study of the EBOVAC-Salone trial were used for the Luminex analysis [23]. Since the IDs of this study were not linked to the malaria study, we had to use a matching algorithm and could only identify a matching serum sample in about 81% of our study participants. Another limitation involved the assessment of clinical episodes of malaria. In Stage 2 of the EBOVAC-Salone trial, adverse events were collected up to 28 days after each vaccine dose, while only serious adverse events were collected throughout the study. This means that episodes of non-serious clinical malaria were not recorded from Day 30 to Day 56, which could have caused misclassification of participants who had a non-serious malaria infection in that time interval. If present, this misclassification likely occurred randomly and independently from the assessment of vaccine immunogenicity. This could have resulted in a dilution of the effect of clinical malaria episodes on vaccine immunogenicity if present [28]. Another limitation of this study is that the diagnosis of clinical episodes of malaria was based on a positive RDT in a population with a high background level of infection. RDT positivity may persist for several weeks after recovery from malaria infection, and this could lead to an overestimation of episodes of clinical malaria. However, the presence of a positive RDT suggests that participants had some exposure to malaria parasites recently even if this was not the cause of their illness. Finally, since the presence of fever was a contraindication to vaccination in the EBOVAC-Salone trial, we were not able to evaluate if the Ebola vaccine regimen is equally immunogenic in subjects with symptomatic malaria infection at vaccination, a situation that could happen outside a clinical trial, especially during mass vaccination in response to an ongoing EVD outbreak.

The strength of this study is the evaluation of the effect of malaria in different age groups because malaria is known to affect children more than older individuals. Moreover, by assessing the effect of previous exposure to malaria at screening and episodes of symptomatic malaria after vaccination, our results complement the results presented previously [18], providing a full picture of the effect of malaria on the immune response to the Ad26.ZEBOV, MVA-BN-Filo Ebola vaccine regimen.

## 5. Conclusions

Overall, the results of this analysis and the results presented in a previous article [18] confirm that there is no indication that malaria substantially affects the immunogenicity of the two-dose Ad26.ZEBOV, MVA-BN-Filo Ebola vaccine regimen, and that this vaccine regimen is suitable for EVD prophylaxis in areas where malaria is highly endemic and where the vaccine may be most needed in the future. However, as the clinical trial could not assess the safety and immunogenicity of the Ebola vaccine regimen in participants with clinical malaria at the time of vaccination, the feasibility of delaying vaccination until recovery in people who have clinical malaria should be considered outside outbreak conditions.

## Figures and Tables

**Figure 1 vaccines-11-01317-f001:**
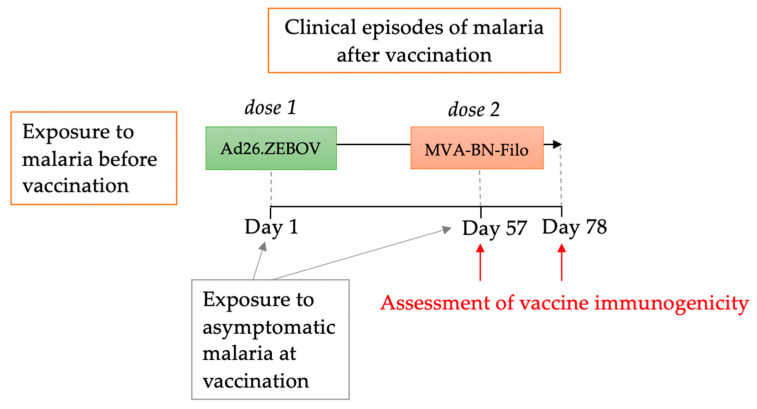
Malaria study conceptual framework. Note: the potential effect of malaria infection on vaccine immunogenicity was considered in three ways: (1) exposure to malaria before vaccination, (2) exposure to malaria at the time of vaccination, described previously [18], and (3) exposure to malaria after vaccination. Assessment of vaccine immunogenicity post dose 1 was evaluated on Day 57. Assessment of vaccine immunogenicity post dose 2 (a measure of the overall immunogenicity of the vaccine regimen), was evaluated on Day 78 (21 days after dose 2 vaccination).

**Figure 2 vaccines-11-01317-f002:**
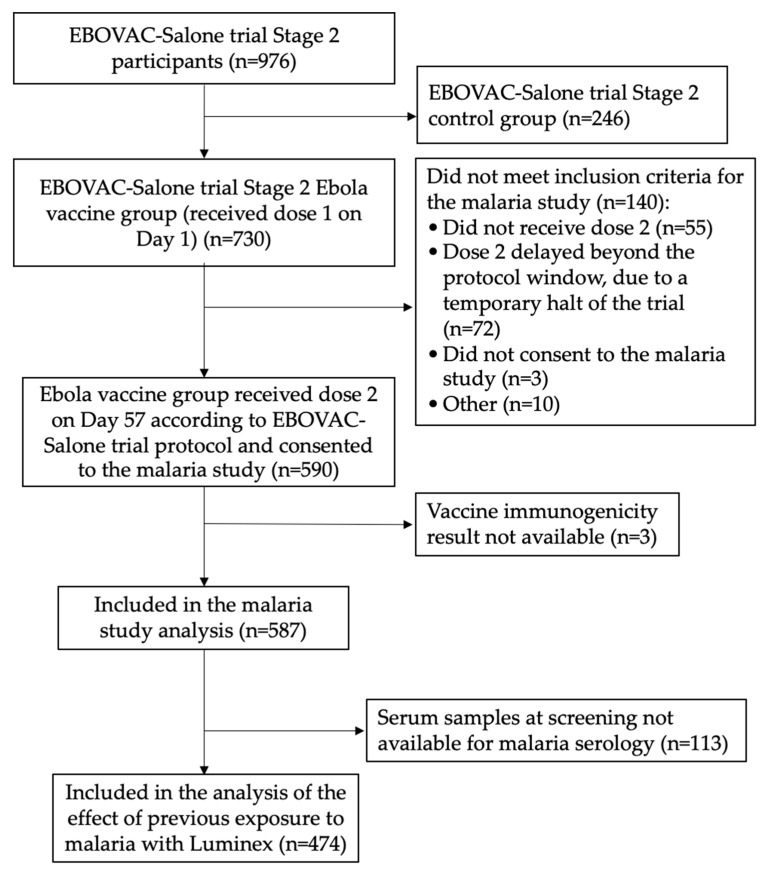
Study flow diagram showing the recruitment process and sample availability for laboratory analysis.

**Table 1 vaccines-11-01317-t001:** Ebola Virus (EBOV) Glycoprotein (GP)-specific binding antibody geometric mean concentrations (GMCs) post dose 1 (measured on Day 57) by categories of previous exposure to malaria, based on participants’ serologic response to a panel of *Plasmodium falciparum* (*P. falciparum)* antigens, indicative of long-term and recent exposure to malaria at the screening visit, overall, and by age cohort.

Long-Term and Recent Exposure to Malaria at Screening	N (%)	Post-Dose 1 EBOV GP-Specific Binding Antibody GMC, EU/mL	GMR ^1^ (95% CI)	*p*
All participants ^2^	N = 474			
Low	144 (30.4)	361 (306–426)	1	0.39
Intermediate	213 (44.9)	324 (283–371)	0.88 (0.72–1.08)	
High	117 (24.7)	402 (337–481)	0.99 (0.78–1.25)	
By age group				
1–3 years	N = 96			
Low	37 (38.5)	783 (612–1002)	1	0.75
Intermediate	28 (29.2)	777 (575–1050)	0.94 (0.65–1.37)	
High	31 (32.3)	694 (491–981)	0.86 (0.55–1.35)	
4–11 years	N = 116			
Low	32 (27.6)	288 (208–400)	1	0.06
Intermediate	56 (48.3)	453 (367–560)	1.35 (0.92–1.98)	
High	28 (24.1)	358 (247–520)	0.89 (0.58–1.37)	
12–17 years	N = 115			
Low	33 (28.7)	365 (263–506)	1	0.83
Intermediate	54 (47.0)	308 (241–393)	0.91 (0.62–1.33)	
High	28 (24.3)	323 (233–447)	0.90 (0.59–1.38)	
≥18 years	N = 147			
Low	42 (28.6)	214 (163–283)	1	0.28
Intermediate	75 (51.0)	189 (152–235)	0.92 (0.67–1.25)	
High	30 (20.4)	314 (230–428)	1.19 (0.85–1.67)	

^1^ Adjusted for baseline EBOV GP-specific antibody concentrations. ^2^ Categories of previous exposure to malaria are age-adjusted. GMR = geometric mean ratio.

**Table 2 vaccines-11-01317-t002:** EBOV GP-specific binding antibody GMCs post-dose 2 (measured on Day 78) by categories of previous exposure to malaria, based on participants’ serologic response to a panel of *P. falciparum* antigens, indicative of long-term and recent exposure to malaria at the screening visit, overall and by age cohort.

Long-Term and Recent Exposure to Malaria at Screening	N (%)	Post-Dose 2 EBOV GP-Specific Binding Antibody GMC, EU/mL	GMR ^1^ (95% CI)	*p*
All participants ^2^	N = 466			
Low	143 (30.7)	8717 (7102–10,699)	1	0.70
Intermediate	206 (44.2)	7927 (6629–9479)	0.94 (0.72–1.23)	
High	117 (25.1)	9331 (7392–11,778)	1.12 (0.82–1.51)	
By age group				
1–3 years	N = 96			
Low	37 (38.5)	23,263 (17,681–30,607)	1	0.90
Intermediate	28 (29.2)	24,544 (17,102–35,225)	1.00 (0.60–1.65)	
High	31 (32.3)	19,313 (10,757–34,676)	0.89 (0.50–1.59)	
4–11 years	N = 115			
Low	32 (27.8)	11,046 (7571–16,116)	1	0.45
Intermediate	55 (47.8)	11,069 (8284–14,791)	1.06 (0.65–1.71)	
High	28 (24.4)	7472 (4794–11,644)	0.76 (0.41–1.42)	
12–17 years	N = 112			
Low	33 (29.5)	8038 (4998–12,926)	1	0.31
Intermediate	51 (45.5)	11,561 (8392–15,927)	1.48 (0.82–2.65)	
High	28 (25.0)	9803 (7338–13,096)	1.21 (0.69–2.12)	
≥18 years	N = 143			
Low	41 (28.7)	3190 (2576–3950)	1	0.06
Intermediate	72 (50.3)	3029 (2372–3869)	0.94 (0.69–1.29)	
High	30 (21.0)	5170 (3676–7271)	1.50 (1.01–2.24)	

^1^ Adjusted for baseline EBOV GP-specific antibody concentrations. ^2^ Categories of previous exposure to malaria are age-adjusted.

**Table 3 vaccines-11-01317-t003:** EBOV GP-specific binding antibody GMCs post dose 1 (measured on Day 57) and clinical malaria episodes, which occurred in between Day 1 and Day 57 ^1^, overall and by age cohort.

Clinical Malaria Post-Dose 1 Vaccination	N (%)	Post-Dose 1 EBOV GP-Specific Binding Antibody GMC, EU/mL	GMR (95% CI)	*p*
All participants	N = 587			
None	412 (70.2)	371 (338–407)	1	0.02
At least one episode	175 (29.8)	323 (275–379)	0.82 (0.69–0.98) ^2^	
By age group				
1–3 years	N = 125			
None	74 (59.2)	750 (630–892)	1	0.23
At least one episode	51 (40.8)	618 (460–830)	0.82 (0.58–1.16)	
4–11 years	N = 133			
None	99 (74.4)	413 (342–498)	1	0.22
At least one episode	34 (25.6)	331 (254–431)	0.80 (0.58–1.11)	
12–17 years	N = 141			
None	120 (85.1)	312 (264–368)	1	0.83
At least one episode	21 (14.9)	327 (209–510)	1.05 (0.65–1.69)	
≥18 years	N = 188			
None	119 (63.3)	260 (220–308)	1	0.05
At least one episode	69 (36.7)	197 (156–249)	0.76 (0.57–1.00)	

^1^ All malaria episodes between Day 1 and Day 29 were recorded (28 days after dose 1 vaccination); between Day 30 and Day 57 only malaria episodes considered serious were recorded. ^2^ Adjusted for age group.

**Table 4 vaccines-11-01317-t004:** EBOV GP-specific binding antibody concentrations post dose 2 (measured on Day 78) and clinical malaria episodes, which occurred in between Day 1 and Day 78 ^1^, overall and by age cohort.

Clinical Malaria Post-Dose 1 and 2 Vaccinations	N (%)	Post-Dose 2 EBOV GP-Specific Binding Antibody GMC, EU/mL	GMR (95% CI)	*p*
All participants	N = 579			
None	350 (60.5)	8489 (7498–9610)	1	0.69
At least one episode	229 (39.5)	9133 (7678–10,863)	1.04 (0.87–1.24) ^2^	
By age group				
1–3 years	N = 125			
None	54 (43.2)	22,601 (18,039–28,317)	1	0.88
At least one episode	71 (56.8)	21,909 (16,020–29,963)	0.97 (0.66–1.43)	
4–11 years	N = 132			
None	92 (69.7)	9470 (7576–11,839)	1	0.24
At least one episode	40 (30.3)	12,062 (8469–17,178)	1.27 (0.84–1.94)	
12–17 years	N = 138			
None	106 (76.8)	9428 (7549–11,775)	1	0.74
At least one episode	32 (23.2)	10,186 (6864–15,117)	1.08 (0.69–1.70)	
≥18 years	N = 184			
None	98 (53.3)	3987 (3275–4852)	1	0.65
At least one episode	86 (46.7)	3741 (3104–4509)	0.94 (0.72–1.23)	

^1^ All malaria episodes between Day 1 and Day 29 (28 days after dose 1 vaccination) and between Day 57 and Day 78 (21 days after dose 2 vaccination) were recorded, and only malaria episodes considered serious were recorded between Day 30 and dose 2 administration (Day 57). ^2^ Adjusted for age group at dose 1 vaccination.

**Table 5 vaccines-11-01317-t005:** Ebola virus glycoprotein binding antibody GMCs post dose 2 (measured on Day 78) and clinical malaria episodes, which occurred after dose 2 in between Day 57 and Day 78, overall and by age cohort.

Clinical Malaria Post-Dose 2 Vaccination	N (%)	Post-Dose 2 EBOV GP-Specific Binding Antibody GMC, EU/mL	GMR (95% CI)	*p*
All participants	N = 579			
None	481 (83.1)	8372 (7511–9333)	1	0.83 ^1^
At least one episode	98 (16.9)	10,775 (8189–14,178)	1.03 (0.79–1.33) ^1^	
By age group				
1–3 years	N = 125			
None	82 (65.6)	24,019 (19,917–28,966)	1	0.29
At least one episode	43 (34.4)	19,117 (11,996–30,467)	0.80 (0.48–1.32)	
4–11 years	N = 132			
None	123 (93.2)	10,051 (8273–12,212)	1	0.59
At least one episode	9 (6.8)	12,299 (5419–27,914)	1.22 (0.53–2.85)	
12–17 years	N = 138			
None	122 (88.4)	9474 (7678–11,689)	1	0.71
At least one episode	16 (11.6)	10,607 (6589–17,076)	1.12 (0.66–1.89)	
≥18 years	N = 184			
None	154 (83.7)	3743 (3217–4356)	1	0.27
At least one episode	30 (16.3)	4591 (3397–6205)	1.23 (0.87–1.72)	

^1^ Adjusted for age group at dose 1 vaccination.

## Data Availability

Following publication, individual deidentified participants’ data and a data dictionary will be made available upon request via the London School of Hygiene & Tropical Medicine research data repository, LSHTM Data Compass at http://datacompass.lshtm.ac.uk (accessed on 31 July 2023). Requests with a defined analysis plan can be sent via LSHTM Data Compass.

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
