# Peer review of "The Effect of Previous Exposure to Malaria Infection and Clinical Malaria Episodes on the Immune Response to the Two-Dose Ad26.ZEBOV, MVA-BN-Filo Ebola Vaccine Regimen"

_vaccines, 2023, doi:10.3390/vaccines11081317_

Round 1
Reviewer 1 Report
This is a very interesting draft publication. There is however, one area that appears to have been overlooked. The baseline ELISA data is not presented. The data appears to exist as seen in the NCT02509494 online content.
It is important to add this because there is a highly significant difference between infants and adults in their titres to GP protein. You need to acknowledge this and discuss. Maybe this is because infants are naive to other Filovirus infections which may be circulating. It is clear that placebo levels were not zero in the online report of NCT02509494 with titres quotes as NA for 1-3 year olds, 42 for 4-11, 74 for 12-17 and 50 for adults over 18. Perhaps there titres represent cross-reactive antibodies raised against other foliviruses such as Ebolavirus Marburg or Ebolavirus Sudan?
You discuss ELISA units in arbitrary Elisa units per ml (EU/ml) which is fine. It is however important to realise that this may not reflect the actual mass of IgG in the sample because it is a function of the avidity and affinity of the antigen specific IgG antibodies present in the sample. Perhaps cross reactive immunity to other filoviruses has suppresses the immune response in adults? The authors must address this secondary observation as it is an obvious omission in the discussion at present.
Apart from this issues which needs to be addressed, this is a concise report with a significant finding.
Author Response
We thank the reviewer for this interesting comment. The reviewer is right that some participants had binding antibodies against the EBOV glycoprotein at baseline and antibody concentration increased with age. A seroprevalence study conducted among participants who attended the screening visit for the Ebola vaccine trial (“EBOVAC-Salone”, NCT02509494) found a baseline seroprevalence of approximately 8% (using a cutoff of >607 ELISA EU/mL, which was calculated previously in an EBOV-naive population in West Africa). However, approximately 53.8% of participants had a result that was above the FANG ELISA lower limit of quantification of 36.11 EU/ml and the concentration of antibodies increased with increasing age. The results of the seroprevalence study have been presented in a previous publication, where we extensively discussed the possible reasons for these findings, including cross-reactivity due to other filovirus infections.1
The Reviewer is also suggesting that a higher concentration of baseline antibodies, which could be the result of cross-reactivity due to other filovirus infections, might explain the lower vaccine-induced antibody concentrations in adults compared to younger age groups. We agree that this is an important question.
A post hoc analysis of the EBOVAC-Salone trial data in adults looked at the correlation between the baseline EBOV GP antibody concentrations and the post dose 2 antibody concentration and found a weak positive correlation (Spearman correlation coefficient: 0.104), which indicated that the baseline EBOV GP antibodies were not associated with a lower concentration of antibodies following vaccination. These results have been previously published in the supplementary material of the paper presenting the results of the EBOVAC-Salone trial in adults.2
To fully answer the reviewer’s question on whether the baseline antibody titres explain the lower antibody concentration post-vaccination, a pooled analysis of individual-level immunogenicity data from different clinical trials of the Ad26.ZEBOV and MVA-BN-Filo Ebola vaccine regimen would probably be needed. This analysis would need to explore not only the effect of baseline antibodies but also other factors such as geographical location, body weight, nutritional status, etc. This analysis would be, however, beyond the scope of the presented study, which aimed to explore the effect of malaria on the immunogenicity of the Ad26.ZEBOV and MVA-BN-Filo vaccine regimen.
References
- Manno D, Ayieko P, Ishola D, et al. Ebola Virus Glycoprotein IgG Seroprevalence in Community Previously Affected by Ebola, Sierra Leone. Emerg Infect Dis 2022;28(3):734-738. (Article) (In English). DOI: https://dx.doi.org/10.3201/eid2803.211496.
- Ishola D, Manno D, Afolabi MO, et al. Safety and long-term immunogenicity of the two-dose heterologous Ad26.ZEBOV and MVA-BN-Filo Ebola vaccine regimen in adults in Sierra Leone: a combined open-label, non-randomised stage 1, and a randomised, double-blind, controlled stage 2 trial. Lancet Infect Dis 2022;22(1):97-109. DOI: 10.1016/S1473-3099(21)00125-0.
Reviewer 2 Report
The study describes effects of previous malaria exposure and clinical malarial episodes on Ebola vaccine immunity. The study is highly relevant in Ebola virus affected countries with high malaria burden.
Major issues:
1. The manuscript has a significant duplication in most parts of the methods and some results (Figure 2, Table 1) with a previous article (10.1093/cid/ciac209) which must be addressed to reduce the plagiarism.
Minor issues:
1. Table 4. Please re-check the Ebola antibody titer in age group 1-3 with no malaria episode, 750 (338-407)- must have a typo.
2. Please clarify how the author adjusted the baseline EBOV GP-specific antibody concentrations.
3. Author should add some discussion how the malaria episode contributed the lower Ebola vaccine antibody in the study subjects.
Author Response
We thank the Reviewer for the insightful and helpful comments. We have taken the comments into consideration and we have revised the manuscript accordingly. Please, find below our point-by-point response.
Major issues:
- The manuscript has a significant duplication in most parts of the methods and some results (Figure 2, Table 1) with a previous article (10.1093/cid/ciac209) which must be addressed to reduce the plagiarism.
Response: thank you for flagging this issue. This happened because we needed to refer to some of the characteristics of the subjects included in the study underlying this publication which had been described in a previous paper focusing on different objectives.1 The Editor made the same comment and highlighted the sentences that needed editing. Consequently, in the revised manuscript, we have rephrased these sentences, moved Table 1 to the supplementary material and we have changed some of the wording in Figure 2. However, some repetitions could not be avoided. For example, the use of Geometric Means Concentration (GMC) and Geometric mean Ratios (GMR) is a standard way of summarising data and is the appropriate method for analysing the type of data presented in our previous publication and in this paper. The number of participants and their distribution by age group is the same as in the previous paper,1 because, as we have explained in the methods, this is the same cohort of participants. Similarly, we did not change the names and descriptions of the vaccines and the malaria antigens used in the Luminex analysis because these are technical terms and we wanted to report them correctly.
Minor issues:
- Table 4. Please re-check the Ebola antibody titer in age group 1-3 with no malaria episode, 750 (338-407)- must have a typo.
Response: thank you for spotting this error. We have corrected the confidence interval, which was reported incorrectly. We have also checked all the results presented in the paper and supplementary material and corrected a few minor rounding errors.
- Please clarify how the author adjusted the baseline EBOV GP-specific antibody concentrations.
Response: we adjusted for the baseline EBOV GP-specific antibody concentrations by including this variable in the multivariable model. We have now made this clear in the methods section (rows 221-223).
- Author should add some discussion how the malaria episode contributed the lower Ebola vaccine antibody in the study subjects.
Response: we have added a sentence in the discussion section (rows 335-337).
Reference
- Ishola D. and the EBOVAC-Salone Malaria Infection Sub-Study Team. Asymptomatic Malaria Infection and the Immune Response to the 2-Dose Ad26.ZEBOV, MVA-BN-Filo Ebola Vaccine Regimen in Adults and Children. Clinical infectious diseases: an official publication of the Infectious Diseases Society of America 2022;75(9):1585-1593. DOI: 10.1093/cid/ciac209.